# Mindfulness-Based IARA Model^®^ Proves Effective to Reduce Stress and Anxiety in Health Care Professionals. A Six-Month Follow-Up Study

**DOI:** 10.3390/ijerph16224421

**Published:** 2019-11-12

**Authors:** Massimiliano Barattucci, Anna Maria Padovan, Ermanno Vitale, Venerando Rapisarda, Tiziana Ramaci, Andrea De Giorgio

**Affiliations:** 1eCampus University, 22060 Novedrate, Italy; massimiliano.barattucci@uniecampus.it; 2Kiara Association, 10125 Torino, Italy; 3Occupational Medicine, Department of Clinical and Experimental Medicine, University of Catania, 95121 Catania, Italy; ermannovitale@gmail.com (E.V.); vrapisarda@unict.it (V.R.); 4Faculty of Human and Social Sciences, University of Enna “Kore”, 94100 Enna, Italy; tiziana.ramaci@unikore.it

**Keywords:** mindfulness, DERS, psychosynthesis, self-awareness, perceived stress, HCPs

## Abstract

Changes in the health care environment, together with specific work-related stressors and the consequences on workers’ health and performance, have led to the implementation of prevention strategies. Among the different approaches, those which are mindfulness-based have been institutionally recommended with an indication provided as to their effectiveness in the management of stress. The aim of the present study was to analyze the efficacy of the mindfulness-based IARA Model^®^ (an Italian acronym translatable into meeting, compliance, responsibility, autonomy) in order to ameliorate perceived stress, anxiety and enhance emotional regulation among health care professionals (HCPs; i.e., doctors, nurses, and healthcare assistants). Four hundred and ninety-seven HCPs, 215 (57.2%) of which were women, were randomly assigned to a mindfulness-based training or control group and agreed to complete questionnaires on emotion regulation difficulties (DERS), anxiety, and perceived stress. Results showed that HCPs who attended the IARA training, compared to the control group, had better emotional regulation, anxiety and stress indices after 6 months from the end of the intervention. Furthermore, the results confirmed the positive relationship between emotional regulation, perceived stress and anxiety. The present study contributes to literature by extending the effectiveness of IARA in improving emotional regulation and well-being in non-clinical samples. Moreover, the study provides support for the idea that some specific emotional regulation processes can be implicated in perceived stress and anxiety. From the application point of view, companies should invest more in stress management intervention, monitoring and training, in order to develop worker skills, emotional self-awareness, and relational resources.

## 1. Introduction

An increasing number of studies have investigated the varied and numerous work-related stress factors that affect health care professionals (HCPs). HCPs are described as being at significant risk of occupational stress and burnout which can also affect, in turn, the relationship with patients [1,2]. Moreover, the psychological issues of HCPs are also due to interactions with patients and caregivers [3], and the emotional exhaustion of HCPs impacts on performance and job satisfaction [4].

Some recent changes in the health care environment (e.g., policy reforms, market dynamics, new technology, decrease in funds, etc.) impact on both the demands on human resources (responsibilities, required quality standards and pressure) and on personal flexibility and adaptation [5].

Furthermore, several organizational aspects have frequently been described as work-related stress factors for HCPs: Role conflicts, unpleasant environmental conditions, administrative work, lack of control over work processes, lack of support, lack of a clear career path, limited autonomy, and poor relations with collaborators [3,6]. The aforementioned negative effects on HCPs entail costs for society (e.g., absence due to sickness, early retirement, costs to the healthcare system [7]), the organization (e.g., need for recruitment and training of new human resources, reduction in clinical hours, absenteeism [8]), and patients (e.g., satisfaction, perceived quality of care, medical errors [6]). 

The pervasive changes to the health care environment, together with specific work-related stressors and the consequences on workers’ health and performance, have led to a major review of personnel management strategies, implementation of prevention strategies and organizational solutions [9,10]. In order to support the organization and HCPs, and to improve their impact on patient’s well-being/health, there is a growing use of training protocols, psycho-social interventions and mindfulness practices which help people to achieve greater self-awareness, and emotional regulation [11]. These approaches use various techniques to improve emotional management such as educational interventions [12], relaxation and breathing techniques [13], imagery [14], interpersonal skills, self-management skills, role-play, psycho-social support, empowerment [15], cognitive behavioral therapy [16], psychodrama, and other various training for psychological skills [17]. In particular, among the different approaches and interventions, those which are mindfulness-based (MBI, MBSR, MBCT, MM, ACT, etc.) have been institutionally recommended [18] with an indication provided as to their effectiveness in the management of psychological indexes and burnout [19] in different populations [20] and in HCPs [12,21,22]. Together, these approaches have been demonstrated to be useful for the purpose of promoting positive relationships at work [9,17,22,23,24].

In particular, effectiveness in reducing perceived stress seems to be linked to the role that some practices have in fostering self-awareness, attention, empathy, cognitive flexibility, compassion, and, more generally, emotional competencies, including acceptance of emotions and emotional regulation [9,10]. The mechanisms of the action of mindfulness seem to be endorsed by emerging evidence that emotional intelligence can act as a moderator of perceived stress [13,15,16,24,25,26].

In order to improve the well-being of HCPs by including both the self-awareness/mindful approaches and the group work that increases its cohesion, IARA training — an Italian acronym translatable into meeting, compliance, responsibility, autonomy — which is a mindfulness-based integrative approach currently emerging in the literature [27,28,29,30], was trialed. IARA is a model that encompass mindfulness, psychosynthesis, and counseling principles using emotional education, role-play, relaxation and breathing techniques, guided imagery, inter-personal and self-management skill improvement.

Previous studies have shown that IARA was effective in different kinds of clinical subjects [27,28,29,30], in modifying several negative psychological indexes: Reducing symptoms of the disease, perceived pain, anxiety, and general psychological impact, while improving the quality of life; however, no study has been done to demonstrate its effectiveness on non-clinical subjects. 

Basing on the indications provided by the literature [8,15,16,23,25], emotional regulation is the key factor in mindfulness-based interventions, which is able to modulate individual responses to stress. Following the above-mentioned rationale and the cited importance of developing intervention tools for stress at work, the study aims at testing the efficacy of IARA training in a sample of workers in reducing emotional difficulties, and perceived stress and anxiety. Moreover, taking into account the results of various studies that highlighted the importance of some socio-work factors in the perception of stress and emotional regulation [25,31], the present research aims to explore the effect of different socio-working conditions on the measured variables.

For this purpose, we designed a pre-post evaluation study with control (T0 and T1), considering the aforementioned variables: Emotional regulation, perceived stress, and anxiety. 

Consequently, we hypothesized the following:

**Hypotheses** **1(H1):**
*IARA training will reduce difficulties in emotional regulation (H1a), perceived stress (H1b) and anxiety (H1c). More specifically, the study assumes that workers which followed the IARA training, compared to colleagues of the control group, will have lower values in emotional difficulties, stress and anxiety, six months after the end of the intervention.*


**Hypotheses** **2(H2):**
*Emotional difficulties are positively correlated with anxiety and perceived stress. The research expects that the higher the worker experiments with difficulties in emotional regulation, the more they will perceive stress and anxiety. Furthermore, the research hypothesizes that some of the six specific dimensions of emotional difficulties should be particularly linked to stress and anxiety.*


**Hypotheses** **3(H3):**
*Demographical variables and work variables affect emotional regulations, anxiety, and stress. More precisely, the study expects that there will be gender, age, education, and seniority differences for all the measured variables.*


## 2. Material and Methods

The research was configured as a randomized pre-post evaluation with a comparison group, which included the completion of a questionnaire at the beginning (T0) and at the end of the training (T1). Baseline assessment was managed in November 2018, while Follow-up in May 2019. All procedures performed in present study were in accordance with the 1964 Helsinki Declaration and its later amendments or comparable ethical standards. All participants gave their signed consent to participate in the study. The study was approved by the local ethics committee.

From different Italian public hospitals, 602 HCP volunteer participants were recruited. The research involved many wards, such as oncology, general medicine, neurology, general surgery, gastroenterology, orthopedics, traumatology, urology, otolaryngology, pulmonology, and home care professionals. All HCPs belonging to these wards—doctors, nurses, and healthcare assistants—were randomly assigned to a control group (N = 301) or to an IARA training program (N = 301). Overall, 497 workers participated in filling out a questionnaire at two different time points (T0, before the training, and T1, 6 months after the end of the training), 284 (57%) of which were women. The final sample was made up of 295 HCPs for the IARA group (response rate = 99%) and of 202 HCPs for the control group (response rate = 68%; Figure 1).

The mean age was 40.35 years (SD = 10.95), while average organizational seniority was 10.98 years (SD = 10.69). One hundred sixty-eight respondents (33.8%) were single, 280 (56.4%) were married/cohabiting and 24 (4.8%) were divorced or widowed (19 missing cases). Thirty-four (6.9%) had a junior high school degree or lower, 212 (42.8%) a high school degree and 231 (46.5%) a university degree (14 missing cases). With reference to employment contracts, 341 (68.7%) held a permanent contract, and 129 (26.1%) had a temporary contract (20 missing cases). 

### 2.1. IARA Meeting Training

Each IARA training group consisted of at least 18 but no more than 22 HCPs. Each group met four times for eight hours. In the first meeting, after a general IARA introduction and an introductory presentation from participants and trainers (all IARA trainers followed a specific qualifying course and were a psychologist and a neuroscientist, a psychologist and nurse, or psychologist and a director of nursing service depending on the training meeting; further information in Table 1), each HCP was invited to present him/herself and share some experiences belonging to daily work life in order to create a group climate. After this, the counseling principles were taught also using the role-play techniques. In the second meeting both oval and star diagrams [32,33] belonging to transpersonal psychosynthesis were explained. Role-play was also used in this meeting session, improving HCP awareness by reflecting on three psychological concepts such as acceptance, listening, unconditional positive acceptance of oneself and others. Finally, during this meeting, a SWOT analysis (i.e., Strengths, Weaknesses, Opportunities and Threats analysis) was presented and explained. In particular, HCPs were invited to pay particular attention to the strength and opportunity elements included in the SWOT.

The third meeting involved education on emotions. In particular, the session involved a deeper exploration of the recognition of primary emotions (astonishment, disgust, fear, anger, joy/happiness, sadness; shame was also considered) and techniques to particularly regulate anger and fear. Moreover, a basic mindfulness exercise was proposed: The attention to breathing and to the emerging thoughts as a first step to improve the awareness of mental activity and to stay in the present moment. During the final meeting, specific guided imagery IARA exercises [34] were explained and demonstrated. Finally, HCPs shared their impressions of the training and some proposals for implementing the IARA in their wards.

### 2.2. Questionnaires

All participants completed questionnaires before the first IARA meeting and six months after the last meeting (as a follow-up measure). 

### 2.3. Anxiety: Zung Self-Rating Anxiety Scale (SAS)

A short self-report scale was used to measure an individual’s anxiety level [35]. Cognitive, autonomic, motor, and central nervous system symptom manifestations were grouped into twenty items. For each item the participants indicated how much each statement applied to them within a period of one or two weeks prior to testing. Each question was scored on a Likert scale ranging from 1 (only a little of the time) to 4 (most of the time). The total raw scores ranged from 20 to 80. Consequently, the raw score must be converted to an anxiety index. The scores fall into four categories: Normal range (20–44); mild to moderate anxiety level (45–59); marked to severe anxiety level (60–74); extreme anxiety level (75–80). Cronbach’s alpha T0 = 0.837; alpha T1 = 0.807.

### 2.4. Difficulties in Emotion Regulation Scale (DERS-36)

The emotion regulation scale is one of the most used tests to assess individuals’ typical levels of emotion dysregulation. Six domains are included: Non-acceptance of negative emotions, inability to engage in goal-directed behaviors when distressed, difficulties controlling impulsive behaviors when distressed, limited access to emotion regulation strategies perceived as effective, lack of emotional awareness, and lack of emotional clarity [36] and it is characterized by 36 items, each item in a Likert scale ranging from 1 (almost never) to 5 (almost always). As such, total scores on the DERS can range from 36 to 180. Cronbach’s alpha T0 = 0.851; alpha T1 = 0.864.

### 2.5. Perceived Stress Scale (PSS)

The perceived stress scale is the most widely used psychological instrument for measuring the perception of stress [37]. The questionnaire has 10 items which were designed to measure how unpredictable, uncontrollable, and overloaded respondents find their lives. Scores are obtained by reversing responses to the four positively stated items and then summing across all scale items. Individual scores range from 0 to 40 with higher scores indicating higher perceived stress. The scores fall into three categories: Low stress (0–13); moderate stress (14–26); high perceived stress (27–40). Cronbach’s alpha T0 = 0.843; alpha T1 = 0.829.

### 2.6. Data Analysis

Data were analyzed using SPSS 24 statistical package (Version 25.0; IBM). Comparability of both groups and randomization success were verified excluding differences between them in terms of socio-working conditions (through χ2) and T0 scores on the three measured factors (through *t*-test).

ANOVA with repeated measures 2 (Group: Training − Control) × 2 (Time: Before − Follow up) was performed to test for group differences on the different time point for all the measured variables, and the differences between pre-training (T0) and follow-up (T1; H1). Moreover, in order to verify the other specific hypothesis (H3), we conducted correlation analyses and *t*-tests.

## 3. Results

χ2 analyses revealed no differences between groups (training and control) for any of the demographic and work characteristics: Gender (χ2 = 0.000, *p* = 1.000), age (χ2 = 0.000, *p* = 1.000), education (χ2 = 0.604, *p* = 0.745), marital state (χ2 = 0.686, *p* = 0.813), and organizational seniority (χ2 = 0.000, *p* = 1.000). 

*t*-test analyses highlighted that training and control groups were almost overlapping in Time 0: mean scores of all the variables have no differences (Table 2).

*t*-test and effect size measure clearly highlighted the differences between the pre and post IARA intervention mean values for all the scales. The differences between groups (training and control) for the follow-up measures were highly significant and showed a medium effect (Table 3). 

Significant reductions in stress, DERS and anxiety were reported six months after the training compared with the pre-intervention period (T0).

In order to further test H1 and explore the main effects and interactions of the three measured variables, a 2 × 2 covariance analysis with repeated measures (time factor) was performed. 

Overall, in line with H1, results showed that six months after participating in IARA groups, HCPs had a global reduction of emotional difficulties, anxiety and stress values, compared to HCPs who had no training (Table 3 and Table 4 and Figure 2, Figure 3 and Figure 4).

Table 5 and Table 6 show the relationships between all collected variables in the two groups: Emotional difficulties are significantly positively correlated with perceived stress and anxiety, generally confirming H2.

In order to explore if some specific DERS subdimensions had an effect on stress or anxiety, a correlational analysis between the different difficulties in emotion regulation subscales, anxiety and perceived stress were performed. Table 7 shows the relation between DERS subscales, perceived stress and anxiety.

Results seem to indicate that especially difficulties with emotional acceptation and awareness are significantly related to stress and anxiety levels.

Finally, *t*-test and correlation analysis results showed no relation between any demographic variable and any of the scales, and no differences for gender, education and job profile. Consequently, H3 was not confirmed.

## 4. Discussion

Literature has already demonstrated the effectiveness of mindfulness-based strategies on stress such as the IARA intervention on different psychological indexes such as pain, symptoms, quality of life and anxiety [28,29,30,32]. However, IARA training effects have been explored only on people with various diseases, and not on workers.

In the present study, we tested the possible impact of the IARA training for/on HCPs psychological indexes, demonstrating its effectiveness. After six months from the end of the IARA activities, HCPs showed a significant reduction in emotional difficulties, anxiety and perceived stress, compared to those workers who did not attend any training. Interestingly, none of the measured workers’ characteristics (gender, profile, education, seniority, and age) were related to anxiety, perceived stress and emotional regulation levels. HCPs are mostly exposed to similar work-related stress factors, and the role of individual differences seems to play a crucial role in modelling subjective responses [38,39,40], which partly explains the recent growing number of stress interventions in the workplace.

Overall, our results confirmed that self-awareness/mindfulness-based interventions have effects on the well-being of workers [8,10,16,17,18,19,20,21,22,23,24,25,41], and that emotional regulation has an undoubted positive relationship with perceived stress and anxiety levels.

The topic of work-related stress and its consequences on workers’ mental health has assumed a fundamental importance at the legal, institutional, organizational and academic levels because of welfare, costs and productivity issues, especially for people working in the health care system [20,21,42]. Anyway, diverse wards represent different environment, conditions and physical and psychological demands for HCPs [31]. Consequently, management is required to monitor controllable organizational, environmental and relational factors (e.g., climate, conflicts, work-load, etc.) that can be related to critical stress situations, in order to program actions to reduce them. Overall, it seems crucial for companies not only to constrain organizational stress variables, but also to explore and improve stress management interventions that can ameliorate the workers’ way of coping with work-related stress factors and with role issues [43,44]. Indeed, stress at work emerges when skills and resources of a single worker fail to help in facing workplace problems, news and change [41]. IARA differs from other mindfulness-based approaches because it uses many tools, (oval and star diagrams; education on awareness and identification of psychological functions which act here and now; breathing; guided imagery in order to improve emotional awareness and thought management through the use of a conscious use of the mind [27]). Compared with eight-week MBSR—one of the most common mindfulness protocols used—IARA has doubled the work hours (16 vs 32 hours) and many role-play activities which seem to be very important in a working context in order to create a positive group atmosphere. Although IARA has many hours of work, it is structured in four meetings, while MBSR is structured in eight meetings. A disadvantage in a workplace, like a hospital ward, is that IARA requires participants to take four full days off work, yet a good organization shift schedule could allow adequate shift covering. Through this training, single HCPs gain more confidence in managing negative emotions through the use of their own totally aware qualities. This sense of autonomy may be useful to improve the sense of self-efficacy and self-esteem [26,27,28,45,46]. HCPs learn to accept themselves and others for their peculiar characteristics, which helps to dissolve conflicts. Moreover, HCPs become more aware of taking care of the patient looking at them in a holistic sense that is not only as a sick patient but also as a person.

Some limitations of this study should be considered in the interpretation of the results and for future studies. Firstly, one major critique is that this training does not effectively reduce anxiety, but attenuate a non-pathological stress among healthcare participants. Moreover, the low response rate of the control group may have led to the exclusion of the perceptions of individuals not sufficiently motivated to complete the questionnaire after a long time. In reverse, all HCPs tolerated the training sessions well, possibly because they were salaried as a normal working day and considered very important to improve professional competencies.

Another limitation was identified in the fact that we considered perceived stress and anxiety as outcome variables, but future research should investigate whether some specific aspects of difficulties in emotional regulation may act as an antecedent factor, or as a mediator. Besides, the anxiety measure used in the present study assessed state rather than trait anxiety, which possibly limits the generalizability of results. Further research should apply psychometrically robust measures of anxiety and perceived work stress for further evaluation [47,48].

Future investigations will certainly have to investigate more deeply the mechanisms of action of the IARA training on emotional regulation that seems to act directly on perceived stress and anxiety levels.

From a practical point of view, this study suggests that companies should invest more in the area of stress management strategies training for HCPs. The ultimate goal of self-awareness/mindfulness-based practices at work, should not be a mere exercise of style for technicians or simple earning opportunities for trainers, but rather an objective with measurable and verifiable measures [41]. The risk that mindfulness interventions, and more generally stress management training protocols, are implemented in a standardized manner and not tailored to the needs of actual workers, should be given due consideration through rigorous research and the adequacy of the operators’ skills [40]. 

The discontinuous and volatile nature of contemporary jobs imposes the obligation to continuously update the practices and strategies, of training processes, and more generally active interventions, with the aim of developing workers individual and interpersonal skills [5], which helps them to achieve greater emotional self-awareness that can be utilized in preventing stress, improving coping abilities and facing problems at work [42].

## 5. Conclusions 

Stress, anxiety and emotional regulation are quite known among HCPs and literature highlights several approaches in order to improve HCP’s quality of life and psychological issues. 

Here we presented an integrative approach named IARA (a training on meeting, compliance, responsibility, autonomy) which was demonstrate able to improve the psychological indexes investigated among HCPs. 

Encompassing mindfulness approach, counselling and psychosyntesis principles, IARA can be considered a holistic approach, because takes care of the whole person. Through the four scheduled meetings, several professionals (nurse, psychologist, neuroscientist, HCPs director) alternate to provide training to operators regarding emotional observation, relaxation, and other stress-related management skills. Hopefully, IARA and other training protocols which help people to achieve greater self-awareness and emotional regulation, should be implemented in healthcare organizations and more extensively institutionally recommended.

## Figures and Tables

**Figure 1 ijerph-16-04421-f001:**
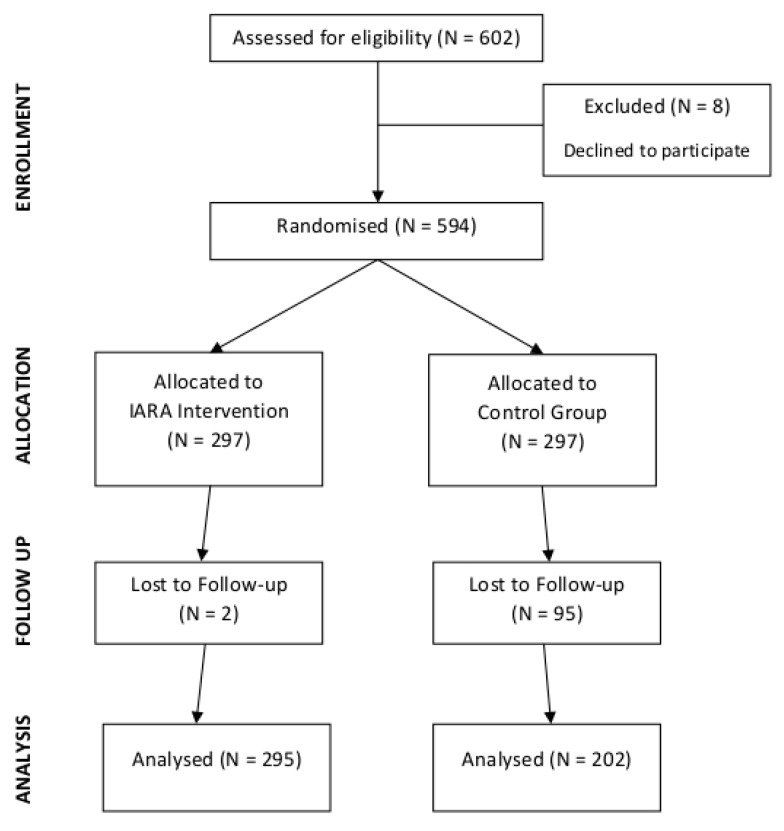
Flow diagram of the progress through the phases of a two-group parallel randomized trial.

**Figure 2 ijerph-16-04421-f002:**
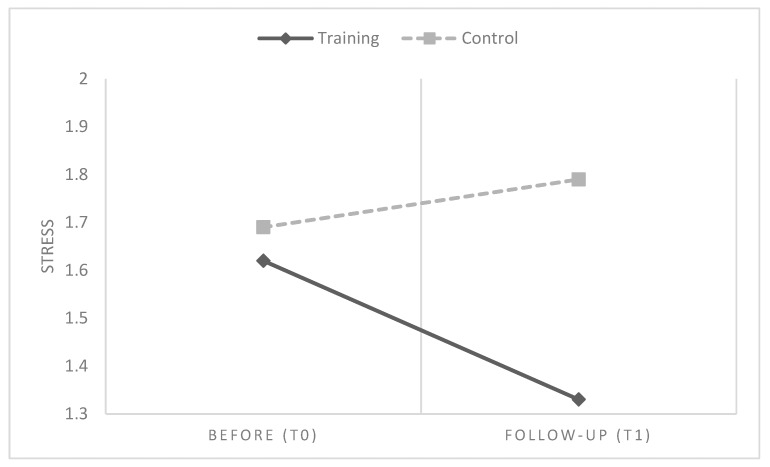
Mean differences between groups and time in perceived stress.

**Figure 3 ijerph-16-04421-f003:**
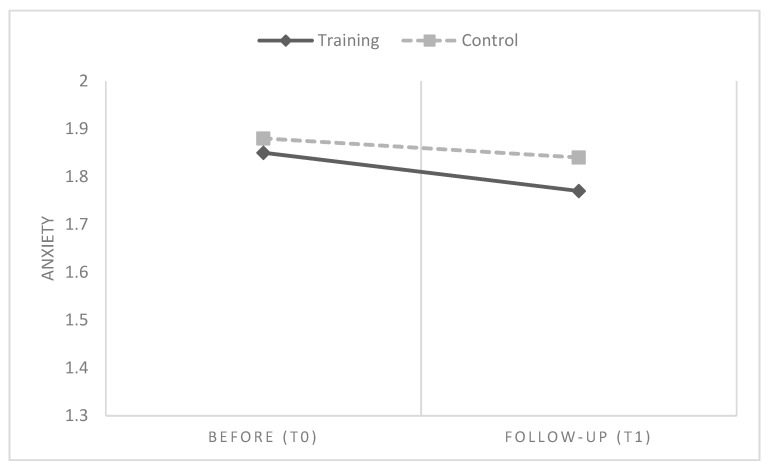
Mean differences between groups and time in anxiety.

**Figure 4 ijerph-16-04421-f004:**
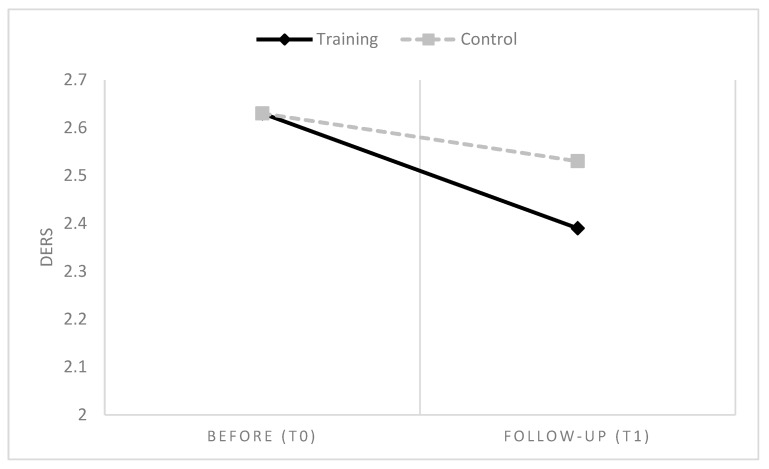
Mean differences between groups and time in DERS.

**Table 1 ijerph-16-04421-t001:** Step by step diagram of the IARA training sessions (HCPs, healthcare professionals; IARA, IARA Model^®^; SWOT, Strengths, Weaknesses, Opportunities and Threats analysis).

**First Meeting: 8 Hours**				
**IARA Trainers**	**Aims**	**Methodology**	**Example**	**Homework**
**Nurse/Psychologist and HCPs director**	Learn IARA principles	taught class	-	-
	create a working group in order to aware the physical, emotional, mental and spiritual levels present in each person	use of open questions filled out individually and then shared with the group	sharing of spare time activities, expectations, desires etc.	increase one’s self-awareness at work and at leisure through continuous questions on the here and now
	develop awareness both of the here and now and positive qualities belonging to everyone	HCPs write at least seven qualities that belong to themselves. When the qualities are fewer, the group helps to reach the seven ones	working group and sharing on qualities; at the end everyone chooses a positive quality of himself and shares it with the group	increase the list of one’s positive qualities, recognize them also in others
	develop the use of the insight as a tool for personality integration	taught class: the importance of the use of words, in particular from “must” to “will”	HCPs share use of their language in a workplace, leisure, families etc.	paying attention to the use of words in daily life
**Second meeting: 8 hours**				
**IARA trainers**	**Aims**	**Methodology**	**Example**	**Homework**
**Nurse/Psychologist/Neuroscientist and HCPs director**	learn the principles and techniques of counseling	use of principles of counseling using clinical case exercises	role-play using principles and techniques of counseling	use principles and techniques of counseling in daily life
	learn how to improve the emotional awareness	taught class: emotions, breathing techniques, and principles of psychosynthesis (Oval and Star diagram)	Breathing exercises; role-play using principles and techniques of psychosynthesis	pay attention to the breath, emotions and try to regulate them thorugh both breath and techniques of psychosynthesis in daily life
	improve the patient-centered care	clinical case exercises	role-play using clinical cases	recognize the relational dynamics and apply personalized assistance to the patient
**T** **hird meeting: 8 hours**				
**IARA trainers**	**Aims**	**Methodology**	**Example**	**Homework**
**Nurse/Psychologist and HCPs director**	learn SWOT analysis	taught class: SWOT analysis	work in small groups applying SWOT analysis in a clinical case	using SWOT analysis in workplace and in daily life
	learn creative imagination	taught class: use of creative imagination in daily life	exercises of creative imagination in a group	using creative imagination in daily life
	learn techniques of the IARA	taught class	use of awareness drawing, qualities and music	using techniques of IARA in daily life
**Fourth meeting: 8 hours**				
**IARA trainers**	**Aims**	**Methodology**	**Example**	**Homework**
**Nurse/Psychologist and HCPs director**	learn the "Seven Psychological Types" according to psychosynthesis	taught class: overview of the "Seven Psychological Types"	work in small groups where each group deepens one of the "Seven Psychological Types"	during daily life pay attention to one’s own typology and that of others
	planning changes in the group through new shared objectives	taught class: methodology to planning changes according to IARA	work in small groups aiming at building new planning	continue work in small group
	improve awareness regarding people’s qualities	excercises on people’s qualities	excercise called "Quality roll"	recognize the qualities in each person during daily life

**Table 2 ijerph-16-04421-t002:** *t*-test for equality (independent samples test) of means by study group (training vs control).

Item	T	*p*	Mean Difference	SE Differences
DERS	−0.35	0.972	0.002	0.05
Anxiety	−1.13	0.262	−0.029	0.02
Stress	−1.25	0.213	−0.064	0.05

**Table 3 ijerph-16-04421-t003:** Mean scores (SE) at the time points and effect sizes for all the measured scales.

IARA		Time 0	Time 1	t Student	Hedge’s *g*
N = 295	DERS	2.63 (0.60)	2.39 (0.32)	3.13 **	0.49
	Anxiety	1.85 (0.27)	1.77 (0.22)	2.32 *	0.32
	Stress	1.63 (0.65)	1.33 (0.50)	6.90 ***	0.51
**Control Group**					
N = 202	DERS	2.63 (0.72)	2.53 (0.73)	1.87	0.13
	Anxiety	1.88 (0.31)	1.84 (0.38)	1.02	0.11
	Stress	1.69 (0.41)	1.79 (0.45)	−1.94	0.18

* *p* < 0.05; ** *p* < 0.01; *** *p* < 0.001.

**Table 4 ijerph-16-04421-t004:** ANOVA 2 (Time) × 2 (Group) results.

Source	df	F	*p*
DERS			
Group	1, 495	3.97	0.046
Time	1, 495	20.45	0.000
Time × Group	1, 495	3.52	0.061
Perceived Stress			
Group	1, 495	54.18	0.000
Time	1, 495	6.99	0.009
Time × Group	1, 495	31.84	0.000
Anxiety			
Group	1, 495	5.59	0.019
Time	1, 495	10.08	0.002
Time × Group	1, 495	.876	0.350

**Table 5 ijerph-16-04421-t005:** Zero-order correlations between the variables of the study in the control group (POST).

Variables	M (SD)	1	2	3	4
1. Age	45.2 (10.27)	-			
2. Seniority	10.21 (9.71.)	0.293 **	-		
3. Perceived Stress	17.2 (7.45)	−0.124	−0.121	-	
4. DERS	10.90(5.5)	0.134	−0.112	0.268 *	-
5. Anxiety	33.81 (8.01)	0.184	0.161	0.318 **	0.232 *

* *p* < 0.05; ** *p* < 0.01; *** *p* < 0.001.

**Table 6 ijerph-16-04421-t006:** Zero-order correlations between the variables of the study in the IARA group (POST).

Title	M (SD)	1	2	3	4
1. Age	44.1 (9.5)	-			
2. Seniority	10.77 (10.31)	0.289 **	-		
3. Perceived Stress	15.75 (6.77)	−0.122	−0.117	-	
4. DERS	11.34 (4.75)	0.129	−0.105	0.282 *	-
5. Anxiety	34.98 (7.95)	0.176	0.151	0.325 **	0.242 *

* *p* < 0.05; ** *p* < 0.01; *** *p* < 0.001.

**Table 7 ijerph-16-04421-t007:** Correlations between DERS subscales, perceived stress and anxiety (POST).

DERS Subscales	Perceived Stress	Anxiety
Acceptation	0.327 **	0.245 *
Distraction	0.284	0.297
Trust	0.242 *	0.224
Control	0.196	0.305 **
Recognition	0.212	0.369 **
Awareness	0.382 ***	0.287 *

* *p* < 0.05; ** *p* < 0.01; *** *p* < 0.001.

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
