# Peer review of "Mindfulness-Based IARA Model® Proves Effective to Reduce Stress and Anxiety in Health Care Professionals. A Six-Month Follow-Up Study"

_ijerph, 2019, doi:10.3390/ijerph16224421_

Round 1

Reviewer 1 Report

The present manuscript aimed to assess the effectiveness of the IARA Model into reduce stress and anxiety among a sample of non-clinical health care professionals. The authors detailed in this manuscript all the steps performed in order to assess 6-month efficacy of the IARA model. One of the strengths of the paper concerns the methodology applied, as well as the sample size.

However, major revisions are needed to make the article suitable for the journal and its readers.

The Abstract could be improved. Please consider a language revision for the statement at page 1 line 16 “The present research [, a pre-post evaluation], aimed to…”, (subject + verb, etc..). Also, capitalize the “HealthCare Professionals”.

Statements required attentions: eg. “we found that emotional regulation could be considered a moderate predictor”. Authors did not provide a moderate-mediation analysis, thus this statement should necessary paraphrased. Abstract results section result to be a little bit generic.

 The background/introduction was well-written and adequately acknowledges the existing findings, research, practices and literature in this field; but it was mainly focused on few published researches. Some sentences are quite long (eg. the first sentence of the Introduction) and redundant: eg: page 1 lines 35-36 with 44-45, as well as page1 line 37 with page 2 line 51. Please address these issue throughout the whole manuscript.

The present introduction could be improved with a review on the main differences between the IARA Model and other mindfulness protocols, etc…. A brief paragraph is needed, highlighting the advances and disadvantages of the IARA Model over other training/approaches etc...

One of the major concerns are on Materials and Methods section of the manuscript.

Design paragraph, as it was presented does not make sense. Author could merge the first part of the Sample with the Design paragraph. Please address it.

It could be useful for the reader to include dates for the Baseline assessment (T0 and not T1) and the Follow-up assessment (T1 or T2), as well as a flow chart of the assignment procedure of the participants to the control/training conditions.

In sample paragraph it was unclear if sample characteristics referred to the whole sample (N=602) or to the final sample (N=295). Please address it.

In the IARA Model paragraph, the model should be clearly presented, for example authors could include a table which address step by step all the training sessions with aims, methodology, homework, examples etc…

Please delete Table 1, and include alpha coefficients in each measure descriptions.

Amos 22 has been used by authors in order to compute...?. Please clarify.

Data analysis paragraph: authors declared to performed ANOVA (rep. meas.), but they reported the chi squared statistic (page 4 line 173) as a first analysis in the result section! Please address it.

Could be useful to report in Table 2, means and SD for each groups at T0 and T1, and compute t-test and hedges g as the effect size measure (instead to the mean differences). Please report the attrition rate from the T0 to the T1.

Please uniform all the table, using the name of the measures in the analysis.

Hp1 has been partially reached. A strong effect has been found only in perceived stress on time and group conditions. The “weak” effect of Group on DERS scores is likely to be attributable to the inflate effect caused by sample size or the multiple comparisons issue. In order to address this issue, could be useful to include an index of power and effect size.

 Concern ANOVA 2x2 it could be useful for the reader if authors describe deeply in which group, time condition were found sign. differences on the Perceived Stress, DERS, and Anxiety measure.

Results descriptions seems to be skinny.

Hp3 has been described at the end of the introduction but not fully addressed in the results section.

Please report zero-order correlations (see Table4) for the control and the training groups separately.

Please delete Table 5. Authors should clearly state the aims of the study. It is not clear because they reported correlation between DERS sub dimensions and perceived stress and anxiety. Please remove or include a specific hypothesis for this analysis.

Again, the Multiple regression showed a weak relationship between Anxiety, Perceived stress and DERS. The model account about the 9% percent. This latter datum best represents an artifact due to the methods/strategies used into perform regression (ie. the case of stepwise bias). Please consider to remove this from the paper.

Despite all the comments mentioned above, I wish to put authors attention to the anxiety measure applied to this study. The anxiety measures used aimed to assess “State anxiety” rather than “trait” and clearly represent a subjective anxiety. This limited strongly the efficacy and the generalizability of the IARA Model.

One major critique could be: this training does not effectively reduce anxiety but attenuate a non-pathological stress among healthcare participants. In fact, no post-treatment assessment has been included in order to detect/control validity threats in baseline-follow-up treatment.    

In addition, authors did not provide a clear datum on the effectiveness of the training among T0-T1, separately for each groups. They only attested that both groups did not different in the baseline condition. I would encourage authors into provide a table with M/SD for each group and each time, as well as t-test and a measure of effect size.

In the limitations paragraph authors could include the necessity into apply psychometrically robust measures of anxiety (eg. the STICSA) in order to avoid a series of methodological issues, se for example:

Balsamo, M., Romanelli, R., Innamorati, M., Ciccarese, G., Carlucci, L., & Saggino, A. (2013). The state-trait anxiety inventory: shadows and lights on its construct validity. Journal of Psychopathology and Behavioral Assessment35(4), 475-486.

Balsamo, M., Cataldi, F., Carlucci, L., & Fairfield, B. (2018). Assessment of anxiety in older adults: a review of self-report measures. Clinical interventions in aging13, 573.

Carlucci, L., Watkins, M. W., Sergi, M. R., Cataldi, F., Saggino, A., & Balsamo, M. (2018). Dimensions of Anxiety, Age, and Gender: Assessing Dimensionality and Measurement Invariance of the State-Trait for Cognitive and Somatic Anxiety (STICSA) in an Italian Sample. Frontiers in psychology9, 2345.

Conclusion paragraph should be necessary revised in spite of the comments made here.

Author Response

Rev. Comm

The Abstract could be improved. Please consider a language revision for the statement at page 1 line 16 “The present research [, a pre-post evaluation], aimed to…”, (subject + verb, etc..). Also, capitalize the “HealthCare Professionals”.

We improved the abstract as requested by reviewer changing from:

The present research, a pre-post evaluation study with a comparison group, aimed to analyze the efficacy of the IARA Model® (IARA)in order to ameliorate perceived stress, anxiety and enhance emotional regulation among […]

to:

The aim of the present study was to analyze the efficacy of the IARA Model® (IARA) in order to ameliorate perceived stress, anxiety and enhance emotional regulation among […]

Rev. Comm

Statements required attentions: eg. “we found that emotional regulation could be considered a moderate predictor”. Authors did not provide a moderate-mediation analysis, thus this statement should necessary paraphrased. Abstract results section results to be a little bit generic.

We modified all the sentence in Abstract from:

Furthermore, we found that emotional regulation could be considered a moderate predictor of perceived stress and anxiety.

to:

Furthermore, results confirmed the positive relationship between emotional regulation, perceived stress and anxiety

Rev. Comm

The background/introduction was well-written and adequately acknowledges the existing findings, research, practices and literature in this field; but it was mainly focused on few published researches. Some sentences are quite long (eg. the first sentence of the Introduction) and redundant: eg: page 1 lines 35-36 with 44-45, as well as page1 line 37 with page 2 line 51. Please address these issue throughout the whole manuscript.

We have changed the following articles (numbers represent previous articles):

Lamothe M, Rondeau É, Malboeuf-Hurtubise C, Duval M, & Sultan S. Outcomes of MBSR or MBSR-based interventions in health care providers: A systematic review with a focus on empathy and emotional competencies. Complementary Therapies in Medicine; 2016, 24:19–28. http://dx.doi.org/10.1016/j.ctim.2015.11.001 Burton A, Burgess C, Dean S, Koutsopoulou GZ, & Hugh-Jones S. How Effective are Mindfulness-Based Interventions for Reducing Stress Among Healthcare Professionals? A Systematic Review and Meta-Analysis. Stress and Health; 2017, 33(1):3-13. DOI:10.1002/smi.2673 Luken M, & Sammons A. Systematic review of mindfulness practice for reducing job burnout. The American Journal of Occupational Therapy; 2016, 70(2):1-10. doi: 10.5014/ajot.2016.016956 Good DJ, Lyddy CJ, Glomb TM, Bono JE, Brown KW, Duffy MK, Baer RA, Brewer JA, & Lazar SW. Contemplating mindfulness at work: an integrative review. Journal of Management; 2015, 42(1):114-142. https://doi.org/10.1177/0149206315617003 Escuriex BF, & Labbé EE. Health care providers’ mindfulness and treatment outcomes: a critical review of the research literature. Mindfulness; 2011, 2(4):242–253. Khoury B, Lecomte T, Fortin G, Masse M, Therien P, Bouchard V, Chapleau MA, Paquin K, & Hofmann SG. Mindfulness-based therapy: a comprehensive meta-analysis. Clinical Psychology Review; 2013, 33:763–771. doi:10.1016/j.cpr.2013.05.005

With these references:

McCarthy C, Bhandari M. Cochrane in CORR®: Preventing Occupational Stress in Healthcare Workers. Clin Orthop Relat Res; 2019, 477:938-944. doi: 10.1097/CORR.0000000000000735.. Martin-Asuero A, Garcia-Banda G. The mindfulness-based stress reduction program (MBSR) reduces stress-related psychological distress in healthcare professionals. Span J Psychol. 2010;13(2):897–905. Ducar DM, Penberthy JK, Schorling JB, Leavell VA, Calland JF. Mindfulness for healthcare providers fosters professional quality of life and mindful attention among emergency medical technicians. Explore (NY); 2019 pii: S1550-8307(19)30443-4. doi: 10.1016/j.explore.2019.07.015. Lamothe M, McDuff P, Pastore YD, Duval M, Sultan S.Developing professional caregivers' empathy and emotional competencies through mindfulness-based stress reduction (MBSR): results of two proof-of-concept studies. BMJ Open; 2018, 8(1):e018421. doi: 10.1136/bmjopen-2017-018421. Strauss C, Gu J, Pitman N, Chapman C, Kuyken W, Whittington A.Evaluation of mindfulness-based cognitive therapy for life and a cognitive behavioural therapy stress-management workshop to improve healthcare staff stress: study protocol for two randomised controlled trial. Trials; 2018, 19:209. doi: 10.1186/s13063-018-2547-1. Cohen-Katz J, Wiley SD, Capuano T, et al. The effects of mindfulness-based stress reduction on nurse stress and burn- out: A qualitative and quantitative study. Hol Nurs Pract; 2005; 19:26–35

We also added new references:

McCarthy C, Bhandari M. Cochrane in CORR®: Preventing Occupational Stress in Healthcare Workers. Clin Orthop Relat Res; 2019, 477:938-944. doi: 10.1097/CORR.0000000000000735.

We are revised redundant sentences throughout the manuscript, in particular.

these paragraph was deleted:

The literature shows that approaches that improve self-awareness have significant effects on many psychological factors, such as anxiety, stress, self-efficacy, job-satisfaction, productivity etc., and has also been successfully used for the purpose of promoting positive relationships at work (9, 16, 22-23).

and is changed in:

Togheter, these approaches have been demonstrated useful for the purpose of promoting positive relationships at work (9, 16, 22-23).

Moreover, the introductory part of the Introduction changes from:

An increasing number of studies have investigated the varied and numerous work-related stress factors that affect Health-Care Professionals (HCPs). HCPs are described as being at significant risk of occupational stress and burnout, because of the emotionally challenging and physically demanding nature of the job (1) […] HCPs are frequently subjected to a high degree of chronic stress, physical and emotional exhaustion, and burnout, also due to interactions with patients and caregivers (5). In particular, emotional exhaustion in HCPs impacts on performance and job satisfaction (6).

to:

An increasing number of studies have investigated the varied and numerous work-related stress factors that affect Health-Care Professionals (HCPs). HCPs are described as being at significant risk of occupational stress and burnout which can also affect, in turn, the relationship with patients (1, 46). Moreover, HCPs psychological issues are also due to interactions with patients and caregivers (5) and HCPs’ emotional exhaustion impacts on performance and job satisfaction (6)

Rev. Comm

The present introduction could be improved with a review on the main differences between the IARA Model and other mindfulness protocols, etc…. A brief paragraph is needed, highlighting the advances and disadvantages of the IARA Model over other training/approaches etc...

We welcome this request. We changed in Introduction from:

This approach is based on Assagioli’s transpersonal psychosynthesis (30) and encompasses emotional education, role-play, relaxation and breathing techniques, visualization and guided imagery, inter-personal and self-management skills improvement.

To:

IARA differs from other mindfulness-based approach because uses more tools, many of which based on Assagioli’s transpersonal psychosynthesis (Oval and Star diagrams; education on awareness and identification of psychological functions which act here and now; breathing; guided imagery in order to improve emotional awareness and thoughts management through the use of a conscious use of the mind

and in discussion from:

In this regard, IARA works through the education on awareness and identification of psychological functions which act here and now, breathing, and guided imagery, in order to improve emotional awareness and thoughts management through the use of a conscious use of the mind; the worker, through this training, gains more confidence in managing negative emotions through the use of their own totally aware qualities, and this sense of autonomy may be useful to improve the sense of self-efficacy and self-esteem (45-46, 26-28).To this purpose, the psychosynthesis tools used (i.e., Oval and Star diagrams) are very important in making clear how unique and complex people are. In agreement with psychosynthesis principles (36), each of us relies differently on the various psychological functions (e.g., sensation; emotion-feeling; imagination; impulse-desire; thought; intuition; will power; self-consciousness) preferring some more strongly than others. Through these tools HCPs learn to accept themselves and others for their peculiar characteristics, which helps to dissolve conflicts. Furthermore, during training, IARA operators received a personalized training so that each of them felt their qualities and potentiality enhanced and recognized. So, doing the operator became more aware of taking care of the patient looking at him in a holistic sense that is not only as a sick patient but also as a person.

to:

In particular, IARA differs from other mindfulness-based approach because uses more tools, many of which based on Assagioli’s transpersonal psychosynthesis (Oval and Star diagrams; education on awareness and identification of psychological functions which act here and now; breathing; guided imagery in order to improve emotional awareness and thoughts management through the use of a conscious use of the mind; 30). For example, compared with 8-week MBSR protocol, one of the most common mindfulness approach used, IARA has double the work hours (16 vs. 32 hours) and many role-play activities which are very important in a working context in order to create a positive group atmosphere. Despite IARA has many hours, MBSR is structured in eight meetings, whereas IARA in four meetings. A disadvantage in a workplace like hospital wards is that IARA needs that HCPs taking four working days to follow it, but a good organization shift schedule allows an adequate shift covering. Through this training, single HCP gains more confidence in managing negative emotions through the use of their own totally aware qualities, and this sense of autonomy may be useful to improve the sense of self-efficacy and self-esteem (45-46, 26-28). HCP learns to accept themselves and others for their peculiar characteristics, which helps to dissolve conflicts. Moreover, HCP becoming more aware of taking care of the patient looking at him in a holistic sense that is not only as a sick patient but also as a person.

Rev. Comm

In the IARA Model paragraph, the model should be clearly presented, for example authors could include a table which address step by step all the training sessions with aims, methodology, homework, examples etc…

We welcome this proposal and a Table 1 was done.

Rev. Comm

One of the major concerns are on Materials and Methods section of the manuscript.

Design paragraph, as it was presented does not make sense. Author could merge the first part of the Sample with the Design paragraph. Please address it.

Thank you for this suggestion. We merged the sections.

Rev. Comm

It could be useful for the reader to include dates for the Baseline assessment (T0 and not T1) and the Follow-up assessment (T1 or T2), as well as a flow chart of the assignment procedure of the participants to the control/training conditions.

We modified Table 3 with your suggestions, and we inserted a new figure with the requested flow chart.

Rev. Comm

In sample paragraph it was unclear if sample characteristics referred to the whole sample (N=602) or to the final sample (N=295). Please address it.

We cleared this point.

Rev. Comm

Please delete Table 1, and include alpha coefficients in each measure descriptions.

Done.

Rev. Comm

Amos 22 has been used by authors in order to compute...?. Please clarify.

We are really sorry for the mistake. We deleted “Amos” phrase.

Rev. Comm

Data analysis paragraph: authors declared to performed ANOVA (rep. meas.), but they reported the chi squared statistic (page 4 line 173) as a first analysis in the result section! Please address it.

Thank you for this advice. The firs result showed T-test for equality (independent samples test) of means by study group (Training vs Control), just as a preliminary requirement for the following analysis.

Rev. Comm

Could be useful to report in Table 2, means and SD for each groups at T0 and T1, and compute t-test and hedges g as the effect size measure (instead to the mean differences). Please report the attrition rate from the T0 to the T1.

Thank you for all the suggestions. We modified Table 3.

Rev. Comm

Please uniform all the table, using the name of the measures in the analysis.

Hp1 has been partially reached. A strong effect has been found only in perceived stress on time and group conditions. The “weak” effect of Group on DERS scores is likely to be attributable to the inflate effect caused by sample size or the multiple comparisons issue. In order to address this issue, could be useful to include an index of power and effect size.

 Concern ANOVA 2x2 it could be useful for the reader if authors describe deeply in which group, time condition were found sign. differences on the Perceived Stress, DERS, and Anxiety measure.

Results descriptions seems to be skinny.

Hp3 has been described at the end of the introduction but not fully addressed in the results section.

Please report zero-order correlations (see Table4) for the control and the training groups separately.

We wish to thank the reviewer for these suggestions: We done all requested things

Rev. Comm

Please delete Table 5. Authors should clearly state the aims of the study. It is not clear because they reported correlation between DERS sub dimensions and perceived stress and anxiety. Please remove or include a specific hypothesis for this analysis.

Modified

Again, the Multiple regression showed a weak relationship between Anxiety, Perceived stress and DERS. The model account about the 9% percent. This latter datum best represents an artifact due to the methods/strategies used into perform regression (ie. the case of stepwise bias). Please consider to remove this from the paper.

Removed

Despite all the comments mentioned above, I wish to put authors attention to the anxiety measure applied to this study. The anxiety measures used aimed to assess “State anxiety” rather than “trait” and clearly represent a subjective anxiety. This limited strongly the efficacy and the generalizability of the IARA Model.

We have added this phrase in Discussion as your suggestion:“One major critique could be: this training does not effectively reduce anxiety but attenuate a non-pathological stress among healthcare participants. In fact, no post-treatment assessment has been included in order to detect/control validity threats in baseline-follow-up treatment”   

Rev. Comm

In addition, authors did not provide a clear datum on the effectiveness of the training among T0-T1, separately for each groups. They only attested that both groups did not different in the baseline condition. I would encourage authors into provide a table with M/SD for each group and each time, as well as t-test and a measure of effect size.

We inserted a new table and result description section

Rev. Comm

In the limitations paragraph authors could include the necessity into apply psychometrically robust measures of anxiety (eg. the STICSA) in order to avoid a series of methodological issues, se for example:

We added two recent references as suggest by reviewer:

Balsamo, M., Cataldi, F., Carlucci, L., & Fairfield, B. (2018). Assessment of anxiety in older adults: a review of self-report measures. Clinical interventions in aging13, 573.

Carlucci, L., Watkins, M. W., Sergi, M. R., Cataldi, F., Saggino, A., & Balsamo, M. (2018). Dimensions of Anxiety, Age, and Gender: Assessing Dimensionality and Measurement Invariance of the State-Trait for Cognitive and Somatic Anxiety (STICSA) in an Italian Sample. Frontiers in psychology9, 2345.

Reviewer 2 Report

Thank you for the opportunity to review the manuscript “The IARA Model® proves effective to reduce stress and anxiety in HCPs. A 6-month follow-up study” for International Journal of Environmental Research and Public Health. This manuscript reports on the efficacy of the IARA Model® (IARA) among healthcare professionals. This was an interesting manuscript, and I enjoyed reviewing it. There is much to like with this paper. The data are unique and suitable to answer the questions raised in this research. Overall, it was thought provoking and enjoyable read. I do, however, some comments on how the manuscript needs to be improved.

Page 2, lines 74-80: Please add additional information about the IARA model. Specifically, what is it and how does it differ from other models. I really liked the use of figures to illustrate the relationship between variables of interest. They would be strengthened by including confidence intervals around the predictions so that we can compare them more accurately. Likewise, confidence intervals should be included when describing the results of the study (pages 4-7).

Author Response

Rev. comm.

Page 2, lines 74-80: Please add additional information about the IARA model. Specifically, what is it and how does it differ from other models.

It was requested also by Reviewer 1; we changed the Discussion from:

This approach is based on Assagioli’s transpersonal psychosynthesis (30) and encompasses emotional education, role-play, relaxation and breathing techniques, visualization and guided imagery, inter-personal and self-management skills improvement.

to:

IARA differs from other mindfulness-based approach because uses more tools, many of which based on Assagioli’s transpersonal psychosynthesis (Oval and Star diagrams; education on awareness and identification of psychological functions which act here and now; breathing; guided imagery in order to improve emotional awareness and thoughts management through the use of a conscious use of the mind

and in discussion from:

In this regard, IARA works through the education on awareness and identification of psychological functions which act here and now, breathing, and guided imagery, in order to improve emotional awareness and thoughts management through the use of a conscious use of the mind; the worker, through this training, gains more confidence in managing negative emotions through the use of their own totally aware qualities, and this sense of autonomy may be useful to improve the sense of self-efficacy and self-esteem (45-46, 26-28).To this purpose, the psychosynthesis tools used (i.e., Oval and Star diagrams) are very important in making clear how unique and complex people are. In agreement with psychosynthesis principles (36), each of us relies differently on the various psychological functions (e.g., sensation; emotion-feeling; imagination; impulse-desire; thought; intuition; will power; self-consciousness) preferring some more strongly than others. Through these tools HCPs learn to accept themselves and others for their peculiar characteristics, which helps to dissolve conflicts. Furthermore, during training, IARA operators received a personalized training so that each of them felt their qualities and potentiality enhanced and recognized. So, doing the operator became more aware of taking care of the patient looking at him in a holistic sense that is not only as a sick patient but also as a person.

to:

In particular, IARA differs from other mindfulness-based approach because uses more tools, many of which based on Assagioli’s transpersonal psychosynthesis (Oval and Star diagrams; education on awareness and identification of psychological functions which act here and now; breathing; guided imagery in order to improve emotional awareness and thoughts management through the use of a conscious use of the mind; 30). For example, compared with 8-week MBSR protocol, one of the most common mindfulness approach used, IARA has double the work hours (16 vs. 32 hours) and many role-play activities which are very important in a working context in order to create a positive group atmosphere. Despite IARA has many hours, MBSR is structured in eight meetings, whereas IARA in four meetings. A disadvantage in a workplace like hospital wards is that IARA needs that HCPs taking four working days to follow it, but a good organization shift schedule allows an adequate shift covering. Through this training, single HCP gains more confidence in managing negative emotions through the use of their own totally aware qualities, and this sense of autonomy may be useful to improve the sense of self-efficacy and self-esteem (45-46, 26-28). HCP learns to accept themselves and others for their peculiar characteristics, which helps to dissolve conflicts. Moreover, HCP becoming more aware of taking care of the patient looking at him in a holistic sense that is not only as a sick patient but also as a person.

Rev. comm.

II really liked the use of figures to illustrate the relationship between variables of interest. They would be strengthened by including confidence intervals around the predictions so that we can compare them more accurately. Likewise, confidence intervals should be included when describing the results of the study (pages 4-7). 

Inserted

Round 2

Reviewer 1 Report

I would like to thank the authors who well took good care of all my previous comments. I am satisfied with the current version.

Only some minor typos should be further paied attention to (eg. page 2 line 86). 

English language and style: minor spell check are required.